# B cell receptor dependent enhancement of dengue virus infection

Chad Gebo[1], Céline S. C. Hardy[1], Benjamin D. McElvany[2], Nancy R. Graham[2], Joseph Q. Lu[1,3], Shima Moradpour[1], Jeffrey R. Currier[4], Heather Friberg[4], Gregory D. Gromowski[4], Stephen J. Thomas[1,3], Gary C. Chan[1], Sean A. Diehl[2], Adam T. Waickman[1,3]*

1 Department of Microbiology and Immunology, State University of New York Upstate Medical University, Syracuse, New York, United States of America, 2 Department of Microbiology and Molecular Genetics, The University of Vermont Larner College of Medicine, Vaccine Testing Center, Burlington, Vermont, United States of America, 3 Global Health Institute, State University of New York Upstate Medical University, Syracuse, New York, United States of America, 4 Viral Diseases Branch, Walter Reed Army Institute of Research, Silver Spring, Maryland, United States of America

* waickmaa@upstate.edu

**Data Availability Statement:** scRNAseq data are publicly available through the Gene Expression Omnibus, accession number GSE276766. All other relevant information are in the manuscript and its supporting information files.

## Abstract

Dengue virus (DENV) is the causative agent of dengue, a mosquito-borne disease that represents a significant and growing public health burden around the world. A unique patho-physiological feature of dengue is immune-mediated enhancement, wherein preexisting immunity elicited by a primary infection can enhance the severity of a subsequent infection by a heterologous DENV serotype. A leading mechanistic explanation for this phenomenon is antibody dependent enhancement (ADE), where sub-neutralizing concentrations of DENV-specific IgG antibodies facilitate entry of DENV into FcγR expressing cells such as monocytes, macrophages, and dendritic cells. Accordingly, this model posits that phagocytic mononuclear cells are the primary reservoir of DENV. However, analysis of samples from individuals experiencing acute DENV infection reveals that B cells are the largest reservoir of infected circulating cells, representing a disconnect in our understanding of immune-mediated DENV tropism. In this study, we demonstrate that the expression of a DENV-specific B cell receptor (BCR) renders cells highly susceptible to DENV infection, with the infection-enhancing activity of the membrane-restricted BCR correlating with the ADE potential of the IgG version of the antibody. In addition, we observed that the frequency of DENV-infectible B cells increases in previously flavivirus-naïve volunteers after a primary DENV infection. These findings suggest that BCR-dependent infection of B cells is a novel mechanism immune-mediated enhancement of DENV-infection.

## Author summary

Dengue virus (DENV) is a mosquito-borne pathogen that infects an estimate 400 million people every year. A unique feature of DENV is immune-mediated enhancement, wherein a first (primary) DENV infection can predispose an individual to a more severe secondary infection. The leading explanation for this phenomenon is antibody dependent enhancement (ADE), wherein antibodies generated in response to a primary DENV infection

**Funding:** Funding for this research was provided by the State of New York and the Department of Defense, Medical Research and Material Command. Additional funding was provided by NIH U01AI141997-06 and P20GM125498 (S.A.D.). The funders had no role in study design, data collection and analysis, decision to publish, or preparation of the manuscript. CG, CSCH, JQL, and ATW received salary support from the State of New York. BDM, NRG, and SAD received salary support from NIH U01AI141997-06 and P20GM125498.

**Competing interests:** The authors have declared that no competing interests exist.

bind but fail to neutralize virions introduced during a secondary infection, thereby allowing DENV to gain entry to Fc-receptor expressing cells. Accordingly, a key prediction of this model is that monocytes, macrophages, and other Fc-receptor expressing phagocytes should be the primary cellular reservoir of DENV during an acute secondary DENV infection. However, it has been noted for decades that B cells are a significant cellular reservoir of DENV, including during secondary DENV infection, representing a disconnect in our understanding of immune-mediated DENV tropism. In our study, we identified DENV-reactive B cell receptors (BCR) as a potent DENV entry receptor. This observation suggests that BCR-dependent infection of DENV-specific B cells may be a complementary mechanism of immune-mediated enhancement of DENV-infection that expands upon and complements existing models of antibody-dependent enhancement.

## Introduction

Dengue virus (DENV) is a prevalent arboviral pathogen that poses a significant global public health burden. Transmitted by the bite of infected *Aedes* family mosquitos, DENV co-circulates as four genetically and antigenically distinct types: DENV-1 to -4. While the majority of DENV infections resolve without the need for medical intervention, dengue can quickly progress in some patients to dengue hemorrhagic fever (DHF) or dengue shock syndrome (DSS) [1–3]. The risk factors associated with progressing to severe dengue are complex and incompletely understood. However, the risk of developing severe disease increases significantly in individuals experiencing secondary/heterologous infections [4,5].

The leading mechanistic explanation for this phenomenon is antibody-dependent enhancement (ADE). This model posits that DENV-specific IgG antibodies elicited by a prior heterotypic DENV infection can opsonize DENV without neutralizing infectivity, facilitating uptake by FcγR bearing phagocytes [6–9]. This mechanism is supported by strong epidemiological data and *in vitro* infection studies and has been a foundational model of dengue immunopathogenesis for over 40 years [10–12]. However, an underappreciated facet of the DENV lifecycle is that B cells are the most frequently infected cell type observed in circulation during acute DENV infection [13–17]. This observation has been substantiated by multiple groups using a wide array of assays including flow cytometry, qRT-PCR, mosquito-inoculation, and scRNAseq [13–17]. While B cells are known to be phagocytic [18,19] their antigen-specific receptors provide an additional, specialized means of internalizing extracellular antigens. Accordingly, we hypothesize that binding of a DENV-specific BCR to DENV could facilitate virion uptake in a manner analogous to ADE.

In this study, we demonstrate that the expression of a DENV-specific B cell receptor renders cells highly susceptible to DENV infection. We posit that this process of BCR-dependent enhancement (BDE) of DENV infection represents an additional mechanism of immune-mediated enhancement which may fill some critical gaps in our current understanding of dengue immunopathogenesis.

## Results

### Expression of a DENV-specific BCR renders cells susceptible to DENV infection

The motivation for this study was the appreciation that the structure of an FcγR receptor engaged with a DENV/IgG complex resembles that of a DENV-specific B cell receptor bound

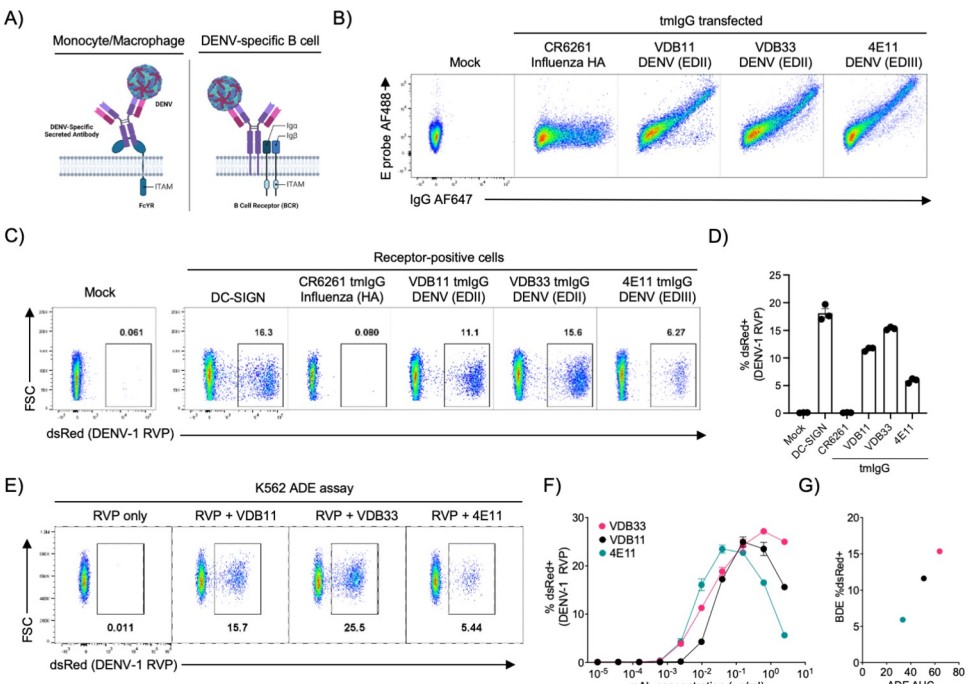

**Fig 1. Expression of DENV-specific BCRs renders cells susceptible to DENV-infection. A)** Schematic representation of DENV/IgG immune complex binding to an FcgR (left), and a DENV virion binding to a DENV-specific BCR (right). Created in BioRender. **B)** Expression and DENV E protein binding activity of 293T cells transfected with the indicated tmIgG constructs. **C)** DENV-1 RVP infection of 293T cells transfected with the indicated tmIgG constructs. DC-SIGN and tmIgG conditions gated on receptor-positive cells. **D)** Quantification of DENV-1 RVP infection of 293T cells transfected with DC-SIGN or the indicated tmIgG constructs. Error bars +/- SEM. Experiments were conducted across three biological replicates in triplicate. **E)** ADE activity of the indicated DENV-specific IgG mAbs at a 2.5ug/ml concentration in K562 cells utilizing a DENV-1 RVP. **F)** Titration of ADE activity of the indicated DENV-specific IgG mAbs in K562 cells utilizing a DENV-1 RVP. **G)** Plot of the ADE and BDE results described above. Error bars +/- SEM.

to DENV (**Fig 1A**). Accordingly, we hypothesized that the expression of a DENV envelope-specific BCR would render a cell susceptible to DENV infection, representing an alternative method of DENV-specific ADE. To test this hypothesis, we generated a panel of transmembrane Ig (tmIg) constructs encoding three well-characterized DENV-specific monoclonal antibodies as well as a control HA-specific tmIg antibody (**Figs 1B and S1**). Cells expressing DENV-specific tmIgG constructs retained their DENV-E protein binding activity, while no DENV-specific binding activity was observed for the HA-specific tmIg construct (**Fig 1B**).

To evaluate the contribution of DENV-specific tmIgG expression on cellular susceptibility to DENV infection, we exposed tmIgG expressing cells to a DENV-1 pseudotyped flavivirus reporter virus particle (RVP) encoding a dsRed fluorescent reporter [20,21], alongside cells transfected with the canonical DENV-entry receptor DC-SIGN (**Fig 1C and 1D**). A reporter virus platform was utilized in this analysis to allow for the accurate discrimination of true cellular infection from tmIgG-mediated binding of virions to the exterior of the target cell, as expression of the dsRed fluorescent reporter encoded by the RVP will only occur following intracellular translation of the RVP genome.

Expression of DENV-specific tmIgG receptors significantly increased susceptibility to infection by the DENV-1 RVP, to levels similar to cells expressing DC-SIGN (**Fig 1C and 1D**). The ability of DENV-specific tmIg constructs to facilitate infection was not limited to IgG isotype BCRs, as a similar pattern was observed in cells expressing tmIgM and tmIgA versions of

DENV-specific antibodies (**S2 Fig**). Notably, the expression of a virus-specific tmIgG is not a universal mechanism for enhancing the susceptibility cells to viral infection, as the expression of HCMV-specific tmIgG constructs did not increase the susceptibility of cells to infection by HCMV. (**S3 Fig**).

While the expression of all DENV envelope-specific tmIgG constructs increased the susceptibility of cells to DENV infection, there was some heterogeneity observed between the constructs. To ascertain if this was due to the specificity of the antibodies for their cognate antigens, we compared the rate of BCR-dependent infection observed in our assay to the rate of ADE achieved with soluble IgG versions of the same mAbs. Strikingly, we observed the same rank-order of ADE and BDE activity of the antibodies when analyzed using a conventional K562 ADE assay (**Fig 1D**, **1E and 1F**). Given that ADE and BDE were in the same rank-order (**Fig 1G**), these results suggest that many of the features of IgG isotype antibodies that are associated with ADE risk may be directly translatable to BDE risk.

## DENV-specific B cells are susceptible to DENV infection

To extend our results showing that transgene-expressed DENV-specific BCRs increase the susceptibility of a permissive cell line to DENV infection, we next examined this in human B cells with endogenous BCR expression. To this end, we leveraged a panel of human memory B cell lines derived from flavivirus-immune donors that were immortalized via genetic reprogramming with BCL-6/BCL-$x_L$ encoding retroviruses and maintained with the T-cell derived stimuli CD40L and IL-21 (**S4 Fig**) [22]. BCL6/BCL-$x_L$-immortalized human memory B cells maintain surface Ig expression, ability to bind their cognate antigen, engage proximal BCR signaling, as well the ability to secrete soluble immunoglobulin [22]. Two cell lines were selected for this analysis: clone 7B9—which expresses an IgG isotype BCR that binds (but poorly neutralizes) DENV-1 to -4 and ZIKV–and clone 2F3 –which expresses a ZIKV-specific IgG isotype BCR but exhibits no DENV-binding activity (**Figs 2A**, **2B**, **2C and S4**).

To determine the impact of BCR specificity on the susceptibility of these cells to DENV infection, both the 7B9 (DENV$^{react}$) and 2F3 (ZIKV$^{react}$) cell lines were exposed to live DENV-2 for 24 hours, followed by extensive washing and additional incubation for 24 hours to allow for viral replication. Cells were then fixed/permeabilized and analyzed by flow cytometry to quantify the abundance of DENV-infected cells within the culture (**S4 Fig**). Consistent with the results obtained with transgenic expression of DENV-specific BCRs, the DENV-specific 7B9 cell line exhibited a significant increase in the frequency of DENV-infected cells relative to the ZIKV-specific 2F3 line (**Fig 2D**). To better characterize the nature of DENV-infection with this B cell system, scRNAseq analysis was performed on both control and DENV-2 exposed 7B9 (DENV$^{react}$) cells (**S5 Fig** and **S1–S4 Tables**). This analysis revealed a similar infection rate (defined as the presence of DENV-2 RNA) within the DENV-2 exposed culture as was observed by flow cytometric analysis of the same culture (**Fig 2E**). Furthermore, both positive- and negative-sense DENV-2 RNA was detected within the cells containing the most significant quantity of DENV-2 RNA, indicating active replication in DENV-specific B cells. (**Fig 2F**). Differential gene expression analysis between DENV-2 infected cells and uninfected cells demonstrated a transcriptional signature consistent with acute lymphocyte activation and cytokine production (**S5 Fig**).

To assess whether primary human B cells are susceptible to DENV infection, we developed a BCR-dependent DENV infection assay that bypasses the need for cells to express a DENV-specific BCR. In this experiment, DENV is bound to an enhancing IgM (VDB33-IgM) and these DENV/Ig immune complexes are crosslinked to the BCR of polyclonal B cells using an anti-IgM secondary antibody (**S6 Fig**) and infection is assessed by intracellular DENV antigen

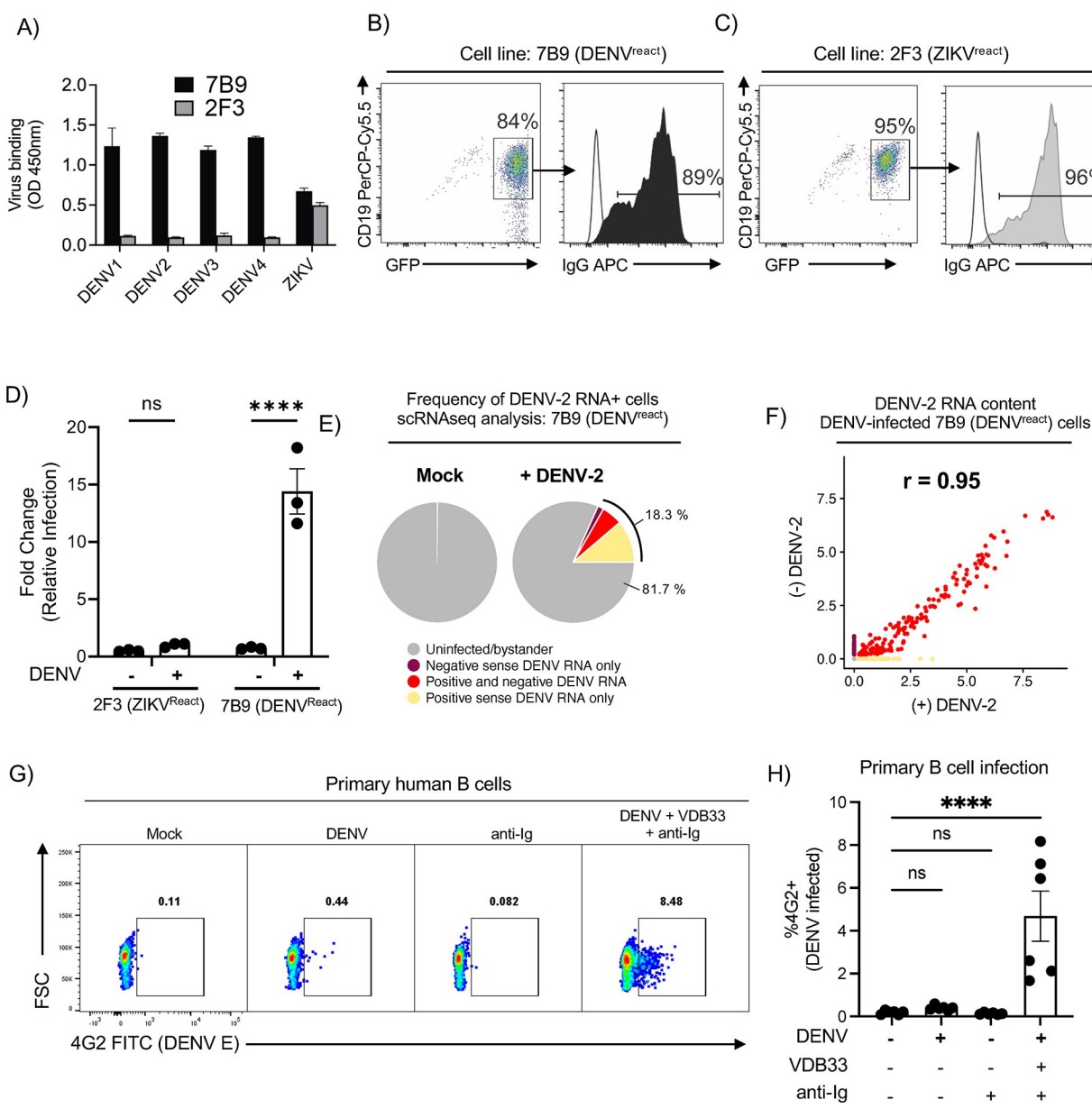

**Fig 2. DENV-specific B cells are susceptible to *ex vivo* DENV infection. A)** Whole virus binding profiles of IgG antibody (1 μg) from cross-reactive 7B9 and ZIKV-specific 2F3 mAb to DENV1-4 and ZIKV by ELISA. **B)** Surface IgG surface expression of immortalized 7B9 and **C)** 2F3 B cells. **D)** Quantification of DENV-2 infection of immortalized 7B9 (DENV^react) and 2F3 (ZIKV^react) B cells at MOI of 10. Error bars +/- SEM. Experiments were conducted across two biological replicates in triplicate **E)** Pie charts of DENV-2 infected 7B9 B cell proportions in control and DENV-2 inoculated conditions as assessed by scRNAseq. **F)** Correlation plot of DENV (+) and DENV (-) sense RNA expression in DENV-2 exposed 7B9 B cells as assessed by scRNAseq. **G)** Representative flow cytometry plot showing the frequency of DENV-4 infected CD19+ B cells under the indicated culture conditions. Detection of DENV-infected cells was performed by staining fixed/permeabilized cells with a FITC-conjugated 4G2 antibody. **H)** Quantification of DENV-4 infected CD19+ B cells under the indicated culture conditions. Error bars +/- SEM. **** p < 0.001, One-way ANOVA with Dunnett's multiple comparisons test, with a single pooled variance. Experiments were conducted across three biological replicates in triplicate.

with the pan-DENV mouse mAb 4G2. We first tested this in Raji cells, which are poorly susceptible to DENV infection even at high MOI yet are highly permissive if transfected with a scavenging receptor such as DC-SIGN [23]. We observed minimal DENV infection in non-transfected Raji cells in all culture conditions except when all components of the DENV/Ig/

BCR crosslinking cocktail were present (**S6 Fig**). To survey the infectibility of primary human B cells we exposed purified B cells from a flavivirus-naïve donor to the same infection conditions described above and assessed the frequency of B cell infection by flow cytometry. As was observed for the Raji cells, significant infection of the primary B cells was only observed in conditions containing the full DENV/Ig/BCR crosslinking cocktail (**Figs 2G**, **2H and S6**).

## Frequency of DENV-infectible B cells increases following primary DENV Infection

A key prediction of the BDE model described above is that the frequency of DENV-infectible B cells in a flavivirus naïve individual should be quite low, but that this frequency should increase following a primary DENV infection due to the accumulation of DENV-specific memory B cells. To address this hypothesis, we analyzed PBMC collected from a dengue human challenge study, wherein flavivirus naïve volunteers were exposed to the attenuated DENV-3 strain CH53489 (**Fig 3A and 3B**). We exposed PBMC collected on days 0, 28, and 90 post DENV-3 infection from 8 volunteers to a DENV-1 RVP, followed by overnight incubation and flow cytometric analysis (**Figs 3A and S7**). We observed a significant increase in the number of DENV-infectible B cells on days 28 and 90 post DENV-3 infection relative to pre-infection (**Fig 3C and 3D**). No such increase in DENV-infectability was observed in either the T cell (**Fig 3E and 3F**) or monocyte compartment (**Fig 3G and 3H**). These results suggest that the accumulation of DENV-specific immunologic memory can alter the susceptibility of an individual's B cells to DENV infection.

## Discussion

In this study, we demonstrate that the expression of a DENV envelope-specific B cell receptor renders cells highly susceptible to DENV infection, and that the frequency of DENV-infectible B cells within a given individual can be modified based on DENV immune status. While an extensive body of literature supports the concept that B cells are a significant circulating reservoir of infectious virus during acute dengue [13–17]–and CD300a has been posited to be a B cell entry receptor for DENV [17]–we believe these are the first results to suggest that B cell susceptibility to DENV infection can be dynamically impacted by prior DENV infections.

Although numerous other viruses exhibit tropism for B cells, we posit that several fundamental features of DENV uniquely position it as amenable to BCR-mediated infection. First, DENV does not have–or require—a single cognate entry receptor. Rather, DENV can utilize a broad range of non-specific scavenging receptors or immunoreceptors to gain access to acidified endosomes of phagocytes, resulting in the viral/endosomal membrane fusion and release of genomic material. Indeed, this is exemplified by prior work which demonstrated that simply tethering DENV to the surface of permissive cells using bi-specific antibodies was sufficient to cause efficient infection, similar in magnitude to what can be achieved via ADE in cells expressing FcγRs [24]. These prior results suggest that the expression of a DENV-specific BCR may render a cell susceptible to DENV infection simply by tethering the virion to the surface of the cell, thereby facilitating viral entry during the process of normal membrane recycling without the need to engage full BCR signaling. However, it is also possible that that BCR-mediated entry of DENV could activate the B cell via multivalent ligation of the BCR complex, thereby increasing protein production and increasing the efficiency of infection in a fashion analogous to what has been described as "intrinsic ADE" for FcγR-mediated entry. Additional work will be required to disentangle these two non-mutually exclusive processes.

An intriguing implication of this model of BCR-mediated entry of DENV is that it may offer some insight into the heterogeneity of risk associated with post-primary DENV

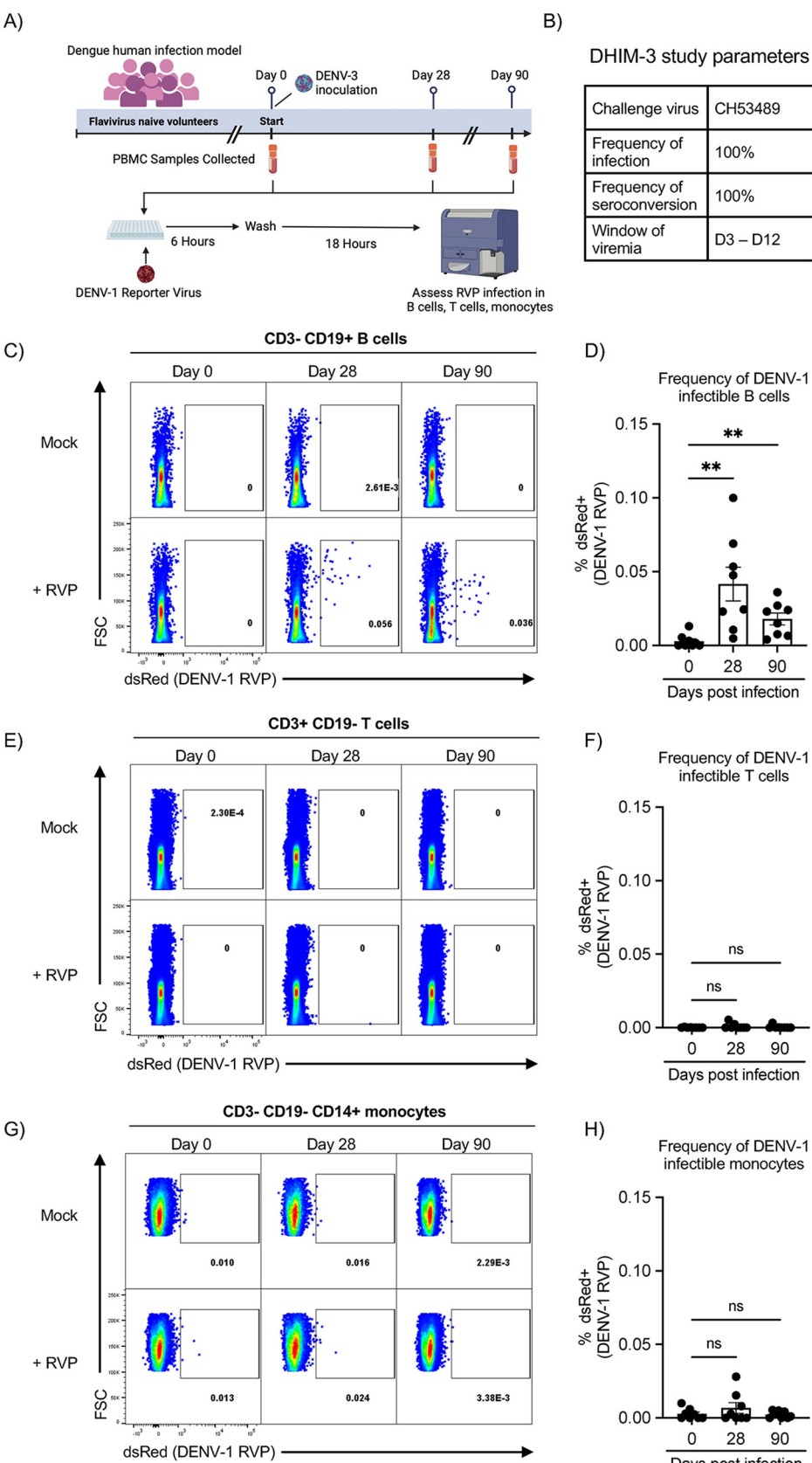

**Fig 3. Frequency of DENV-infectible B cells increases following primary DENV Infection. A)** Schematic representation of the dengue human infection model and *in vitro* DENV-1 RVP infection assay. Created in BioRender. **B)** Key DHIM study characteristic and performance parameters. **C)** Representative flow cytometry plots showing the frequency of DENV-1 RVP infectible viable B cells (CD3-CD19+) in PBMC samples obtained 0-, 28-, and 90-days post DENV-3 infection. **D)** Quantification of DENV-1 RVP infectible B cells from all subjects included in this analysis. **E)** Representative flow cytometry plots showing the frequency of DENV-1 RVP infectible viable T cells (CD3+CD19-) in PBMC samples obtained 0-, 28-, and 90-days post DENV-3 infection. **F)** Quantification of DENV-1 RVP infectible T cells from all subjects included in this analysis. **G)** Representative flow cytometry plots showing the frequency of DENV-1 RVP infectible viable monocytes (CD3-CD19-CD14+) in PBMC samples obtained 0-, 28-, and 90-days post DENV-3 infection. **H)** Quantification of DENV-1 RVP infectible monocytes from all subjects included in this analysis. Error bars +/- SEM. ** $p < 0.01$, paired one-way ANOVA with correction for multiple comparisons (Friedman test with Dunn's multiple comparisons test).

exposures. Increased risk for severe disease following a secondary, heterotypic infection has been associated with a narrow range of low level preexisting DENV antibody titers–but only a small fraction of these individuals actually develop severe disease [5,25,26]. Given the extraordinary stability of memory B cells [27], one potential immunopathological mechanism contributing to severe dengue disease may be the relative abundance of DENV-specific antibodies and DENV virion-specific memory B cells circulating after a primary DENV infection. This may result in a situation where the population of highly-susceptible DENV-specific memory B cells are protected from infection with DENV until neutralizing antibody titers wane below a protective threshold. Accordingly, while the frequency of DENV-specific/DENV-infectible B cells identified in our analysis appears to be greatest 28 days after a primary DENV infection, the high neutralizing antibody titers present at this time point make it unlikely that a secondary infection event could occur. Additional analyses are required to determine if–and how– this mechanism may contribute to risk or protection from DENV infection.

## Materials and methods

### Ethics statement

The DENV-3 human challenge study and all associated analysis was approved by the State University of New York Upstate Medical University (SUNY-UMU) and the Department of Defense's Human Research Protection Office. The investigators have adhered to the policies for protection of human participants as prescribed in AR 70–25. The DENV and ZIKV specific cell lines were obtained from separate studies approved by the Johns Hopkins University Institutional Review Board (IRB) and University of Vermont IRB, respectively. Written informed consent was obtained from volunteers enrolled in all studies.

### Cells and cell lines

PBMC for flow cytometric analysis were obtained from a previously described phase 1, open-label dengue human challenge study [28]. This DENV-3 human challenge study and all associated analysis was approved by the State University of New York Upstate Medical University (SUNY-UMU) and the Department of Defense's Human Research Protection Office. 293T cell line were maintained DMEM media (Gibco, # 11965092) supplemented to 10% FBS, 1% penicillin/streptomycin, and 1% L-glutamine. K562, and U937 DC-SIGN cell line RPMI. Raji cell line was purchased through ATCC and maintained in complete IMDM media (Gibco, # 12440053).

### Generation of immortalized B cell lines

7B9 cells were generated from a flavivirus-naïve volunteer who was vaccinated with DENV1Δ30 and challenged 270 days later with DENV2Δ30 Tonga (NCT02392325). B cells

were isolated 90 days following DENV2Δ30 challenge. This study was approved by the Johns Hopkins University Institutional Review Board (IRB). 2F3 cells were obtained from a DENV-exposed volunteer (DT165) about 1 year following a secondary ZIKV infection (approved by University of Vermont IRB). Written informed consent was obtained from volunteers enrolled in both studies. Both 7B9 and 2F3 cells were generated by transduction of IgM–CD27+ memory B cells with a retrovirus encoding human B cell lymphoma (BCL)-6 and BCL-xL and enhanced green fluorescent protein (GFP) and culture on CD40L-expressing L cell fibroblasts and rhIL-21 (25 ng/mL, Peprotech) as described [23]. Surface IgG expression was performed on live B cells (near-IR fluorescent reactive dye, Invitrogen, cat no. L34975A), that were positive for GFP and CD19-PerCP.Cy5.5 (Clone HIB-19, Biolegend, #302230) using APC anti-human IgG Fc (Biolegend, cat. no. 410712) on a Beckman CytoFlex flow cytometer.

## Viruses and RVPs

DENV-1 (strain Wp74) dsRed Reporter virus (RVP) was prepared as previously described [20,21]. RVPs were titrated on U937 DC-SIGN cells, infectious units (IU) calculated based on the dilution of virus needed to infect 10–15% of cells as previously described [29]. DENV-4 (strain D4-1036) stock was prepared by propagating a low-passage inoculum in C6/36 cells. DENV-2 (strain Thailand/NGS-C/1944) stock was prepared by propagating a low-passage inoculum in Vero cells. RVP infection assays were performed using an approximate MOI of 0.1, with cells incubated with the RVP for 6hrs, followed by and an additional 18hrs of incubation prior to analysis by flow cytometry. Surface staining was performed in PBS + 2% FBS using the reagents and dilutions shown in S5 Table. Aqua Live/Dead (ThermoFisher, L34957) or ZombieUV (Biolegend, 423107) was used for live/dead cell discrimination. Data collection was performed on a BD LSRII or Fortessa flow cytometer and analyzed using FlowJo v10.2 (Becton Dickinson).

## K562 ADE assay

ADE assays were performed using K562 cells as previously described [30,31]. In short, serial dilutions of monoclonal antibodies were incubated with the DENV-1 RVP (in sufficient amounts to infect 10%–15% of U937-DC-SIGN cells) at a 1:1 ratio for 1 h at 37°C. This mixture was then added to a 96-well plate containing $5 \times 10^4$ K562 cells per well in duplicate. Cells were cultured for 18–20 hr overnight in a 37°C, 5% CO2, humidified incubator, follow by analysis by flow cytometry.

## DENV-specific ELISA and neutralization assays

These assays were performed as previously described [32]. Briefly, virus was captured by plate-adsorbed mouse cross-reactive anti-DENV envelope (E) protein monoclonal antibody 4G2. IgG-containing cell culture supernatant was then added and positive DENV binding was detected by alkaline phosphatase-conjugated goat anti-human IgG (Fc) antibody (Millipore Sigma) and p-nitrophenyl phosphate substrate (Millipore Sigma). Reaction color change, indicating DENV-binding, was measured by spectrophotometry as $OD_{405}$. Serum neutralizing antibody titers against DENV1-4 and ZIKV were determined by plaque reduction neutralization test (PRNT), using lowest serum dilution that yielded a 50% reduction in viral plaques ($PRNT_{50}$) as previously described [33,34].

## Generation and expression of transmembrane antibody constructs

Sequences for each monoclonal antibody were appended with an Ig transmembrane domain appropriate to the mAb isotype and cloned into pcDNA3.1(+) expression constructs.

Transient expression of all tmIg constructs was performed in 293T cells using Lipofectamine (Invitrogen, L3000001). Cells were incubated for 18hrs before use in binding and infection assays. Antigen-specificity of the DENV-specific BCR was assessed by staining with biotin-labeled DENV-2 E protein pre-incubated with AF488-tagged streptavidin.

## hCMV infection

hCMV strain TB40E-GFP [35] was diluted in complete RPMI media then incubated with transfected cells at an MOI of 1 at 37C for 6 hours. Cells were then washed before incubating at 37C for an additional 18 hours before quantification of infection by flow cytometry.

## Single cell RNA sequencing

Samples were prepared for single-cell RNA sequencing according to the 10x Genomics RNA-seq protocol. Cells were resuspended at a concentration of 1400 cells/uL in PBS and loaded for a target of 6000–9000 cells per reaction. Cells were loaded for Gel emulsion bead (GEM) generation and barcoding. Construction of 5' gene expression libraries was performed using the Next GEM Single Cell 5' reagent kit, Library Construction Kit, and i7 Multiplex Kit (10x Genomics, CA). was used for reverse transcription, complementary DNA amplification and construction of gene expression libraries. The quality of gene expression libraries was assessed using an Agilent 4200 TapeStation with High Sensitivity D5000 ScreenTape Assay and Qubit Fluorometer (Thermo Fisher Scientific) according to the manufacturer's recommendations. Sequencing of 5' gene expression libraries was performed on an Illumina NextSeq 2000 (Illumina) using P3 reagent kits (100 cycles). Parameters for sequencing were set at 26 cycles for Read1, 10 cycles for Index1, and 90 cycles for Read2.

## 5' gene expression analysis and visualization

Gene expression alignment, sample demultiplexing, alignment, and barcode/UMI filtering was performed using the Cell Ranger software package (10x Genomics, CA) and bcl2fastq (Illumina, CA) using the commands mkfastq and count. The human reference genome (Ensembl GRCh38.93) was combined with the DENV2 genome as an additional chromosome (NC 001474.2). Sequenced transcripts were aligned to a human reference library created using the Cell Ranger mkref command, combined human and DENV2 reference genome, and custom Ensembl GRCh38 DENV2 GTF. Reads were mapped to both the positive and negative sense DENV2 genome. Multi-sample integration, data normalization, dimensional reduction, visualization and differential gene expression were performed in R studio (v4.3.2) using R package Seurat (v4.4.0). The datasetfiltered to exclude genes expressed in fewer than 3 cells, and to contain cells with less than 22% mitochondrial RNA content and between 500–7,500 unique features. The resulting dataset was normalized and scaled using the Seurat functions NormalizeData(), ScaleData(), and FindVariableFeatures(). Data was normalized and scaled, and principal component analysis was performed using RunPCA(). Clustering was based on the first 10 principal components and a resolution parameter of 0.1 using FindNeighbors() and FindClusters(), respectively.

Cells labelled as infected were identified by the presence of both DENV (+) and (-) RNA with expression of both transcripts >0. Differentially expressed genes were identified by applying the FindMarkers() command and the Wilcoxon rank-sum test to the normalized gene expression dataset. A default minimum logFC value of 0.1 and min.pct of 0.01 were used. Genes with a corrected p-value by Bonferroni correction of <0.05 were considered significant. Ingenuity Pathway Analysis (IPA, Qiagen) was performed using genes with a corrected p-value <0.01 and log fold change <-0.25 or >0.25.

### DENV/Ig/BCR crosslinking infection

DENV/antibody immune complexes were generated by mixing DENV-4 ($6.25 \times 10^7$ IU/mL) using an MOI of 62 with 1ug/mL of VDB33-IgM and anti-IgM in 96-well plate. $5 \times 10^4$ Raji cells or primary B cells isolated from flavivirus-naïve healthy donors using MojoSort Human Pan B cell Isolation Kit (BioLegend, 480082) were then added and incubated for 24 hours. Supernatants were harvest at 24hrs for cytokine analysis, while cells were washed and incubated an additional 24 hours before analysis. Cells were intracellularly stained with FITC-labelled DENV E-reactive monoclonal 4G2 antibody at 4μg/mL to quantify DENV infection.

### Statistical analysis

Statistical analyses were performed using GraphPad Prism 9 (La Jolla, CA) with a p-value $< 0.05$ considered significant.

## Supporting information

**S1 Fig. Gating scheme and DENV-1 RVP infection of tmIgG transfected 293T cells. A)** Expression and gating of tmIgG and DC-SIGN in transfected 293T cells **B)** Representative flow cytometry plots showing the frequency of DENV-1 RVP infected cells within the receptor-negative gate of tmIgG and DC-SIGN in transfected 293T cells 24hrs after RVP exposure **C)** Quantification of DENV-1 RVP infected cells within the receptor-negative gate of tmIgG and DC-SIGN in transfected 293T cells 24hrs after RVP exposure
(PDF)

**S2 Fig. Gating scheme and DENV-1 RVP infection of tmIgM and tmIgA transfected 293T cells. A)** Expression and DENV E protein binding activity of VDB33 tmIgM and tmIgA expression constructs. **B)** Expression and gating of tmIgM and tmIgA expression in transiently-transfected 293T cells **C)** Representative flow cytometry plots showing the frequency of DENV-1 RVP infected cells within the DC-SIGN, CR261 tmIgG, VDB33 tmIgA, VDB33 tmIgG, and VDB33 tmIgG positive 293T cells 24 hours after infection **D)** Quantification of DENV-1 RVP infected cells within the DC-SIGN, CR261 tmIgG, VDB33 tmIgA, VDB33 tmIgG, and VDB33 tmIgG positive 293T cells 24 hours after infection
(PDF)

**S3 Fig. Expression of HCMV specific tmIgG. A)** Binding of the indicated IgG isotype mAbs to recombinant/purified HCMV gH protein as quantified by ELISA. **B)** Expression and gating of tmIgG and PDGFRa expression in transiently-transfected 293T cells **C)** Representative flow cytometry plots showing the frequency of HCMV SV40-GFP infected cells within the receptor-positive gate of tmIgG and PDGFRa transfected 293T cells 24hrs after virus inoculation. Cells infected at an MOI of 1 **D)** Quantification of HCMV SV40-GFP infected cells within the receptor-negative gate of tmIgG and PDGFRa in transfected 293T cells 24hrs after virus inoculation
(PDF)

**S4 Fig. Characterization and DENV-infection of 7B9 and 2F3 B cell lines. A)** Schematic representation of 7B9 and 2F3 cell line generation and maintenance. Created in BioRender. **B)** Virus binding ELISA data for mAbs expressed by 7B9 and **C)** 2F3 cell lines. **D)** DENV and ZIKV neutralization profiles of mAb expressed by cell line 7B9. **E)** Gating scheme for flow cytometry analysis of DENV-infected 7B9 and 2F3 cell lines. **F)** Representative flow cytometry plots showing the frequency of DENV-infected 7B9 and 2F3 cell lines after DENV-2 exposure.
(PDF)

**S5 Fig. Quality control metrics and integrated UMAP projections of scRNA seq data. A)** Violin plots indicating number of features (nFeatures) and percentage mitochondrial content (percent mt) pre-filtering of scRNAseq data. **B)** Integrated UMAP projection of cells derived from all conditions prior to subsetting for removal of CD40 ligand positive feeder cell population, and **C)** Feature plot indicating CD40 ligand (CD40LG) expression, highlighting the feeder cell population (cluster 3) which was removed for subsequent analysis. **D)** UMAP projections of scRNAseq data indicating Seurat clusters, **E)** Sample origin, and **F)** Cell cycle phase. **G)** Feature plot indicating DENV (+) and (-) sense RNA expression. **H)** Imputed cell labelling of infected (positive for both DENV positive and negative sense RNA) and uninfected control/bystander cell populations. **I)** Dot plot highlighting selectively upregulated and downregulated DEGs in infected (expressing both DENV (+) and (-) RNA) compared to uninfected control cells (lacking both DENV (+) and (-) RNA expression). Average expression and percent of cells expressing a given transcript are indicated. **J)** Ingenuity Pathway Analysis (IPA) Graphical summary of predicted pathway enrichment in infected cells compared to uninfected controls based on differential gene expression between infected (expressing both DENV (+) and (-) RNA) compared to uninfected control cells (lacking both DENV (+) and (-) RNA expression).
(PDF)

**S6 Fig. DENV infection of polyclonal B cells by BCR/DENV cross-linking. A)** Schematic representation of DENV/Ig/BCR crosslinking assay. Created in BioRender. **B)** Representative flow cytometry plot showing the frequency of DENV-4 infected Raji cells under the indicated culture conditions. Detection of DENV-infected cells was performed by staining fixed/permeabilized cells with a FITC-conjugated 4G2 antibody. **C)** Quantification of DENV-4 infected Raji cells under the indicated culture conditions. **D)** Gating scheme for B cell infection analysis utilizing the DENV/Ig/BCR crosslinking assay.
(PDF)

**S7 Fig. Gating scheme for DHIM-3 PBMC analysis.**
(PDF)

**S1 Table. Quality control metrics for scRNAseq data.**
(DOCX)

**S2 Table. Infected cell characteristics across culture conditions.**
(DOCX)

**S3 Table. Differentially expressed genes, DENV-2 infected cells and uninfected mock cells.**
(DOCX)

**S4 Table. Differentially expressed pathways between DENV-2 infected 7B9 cells and uninfected mock cells.**
(DOCX)

**S5 Table. Reagents for flow cytometry analysis.**
(DOCX)

**S1 Data. Raw data files for figures.**
(XLSX)

**S2 Data. List of expression constructs.**
(DOCX)

## Acknowledgments

We gratefully acknowledge excellent technical assistance provided by Lisa Phelps of the SUNY Upstate Medical University Flow Cytometry Core. We also wish to thank Ted Pierson and Alan Rothman for the constructs and protocols for reporter virus production, and Saravanan Thangamani for the DENV-2 utilized in this study. We acknowledge the Harry Hood Bassett Flow Cytometry and Small Particles Detection facility (RRID:SCR_022147) at the University of Vermont Larner College of Medicine. Clinical trial NCT02392325 was funded by the Bill and Melinda Gates Foundation (OPP1109415).

## Author Contributions

**Conceptualization:** Chad Gebo, Jeffrey R. Currier, Heather Friberg, Gregory D. Gromowski, Stephen J. Thomas, Gary C. Chan, Sean A. Diehl, Adam T. Waickman.

**Data curation:** Chad Gebo, Céline S. C. Hardy, Adam T. Waickman.

**Formal analysis:** Chad Gebo, Céline S. C. Hardy, Adam T. Waickman.

**Funding acquisition:** Stephen J. Thomas, Sean A. Diehl, Adam T. Waickman.

**Investigation:** Chad Gebo, Benjamin D. McElvany, Nancy R. Graham, Joseph Q. Lu, Shima Moradpour, Stephen J. Thomas, Sean A. Diehl, Adam T. Waickman.

**Methodology:** Chad Gebo, Benjamin D. McElvany, Nancy R. Graham, Joseph Q. Lu, Shima Moradpour, Sean A. Diehl, Adam T. Waickman.

**Project administration:** Sean A. Diehl, Adam T. Waickman.

**Resources:** Jeffrey R. Currier, Heather Friberg, Gregory D. Gromowski, Gary C. Chan, Sean A. Diehl, Adam T. Waickman.

**Supervision:** Sean A. Diehl, Adam T. Waickman.

**Validation:** Chad Gebo, Céline S. C. Hardy, Adam T. Waickman.

**Visualization:** Chad Gebo, Sean A. Diehl, Adam T. Waickman.

**Writing – original draft:** Chad Gebo, Adam T. Waickman.

**Writing – review & editing:** Chad Gebo, Céline S. C. Hardy, Benjamin D. McElvany, Nancy R. Graham, Shima Moradpour, Jeffrey R. Currier, Heather Friberg, Gregory D. Gromowski, Stephen J. Thomas, Gary C. Chan, Sean A. Diehl, Adam T. Waickman.

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
