## [Decision Letter · Decision Letter 0]

13 Aug 2024

Dear Dr Waickman,

Thank you very much for submitting your manuscript "B cell receptor dependent enhancement of dengue virus infection" for consideration at PLOS Pathogens. As with all papers reviewed by the journal, your manuscript was reviewed by members of the editorial board and by several independent reviewers. In light of the reviews (below this email), we would like to invite the resubmission of a significantly-revised version that takes into account the reviewers' comments.

To summarize, the manuscript is being returned with three reviews. The reviewers agreed that the study is of substantial interest. However, each reviewer identified key concerns with the current manuscript. While all reviewer concerns should be addressed in a revised submission, the authors should pay particular attention to the following key points:

1. Concerns were raised by multiple reviewers regarding the capacity of the BCR to function as a cell entry receptor for DENV, including the role of CD79a/b.

2. Concerns were raised about the experiments using PBMCs, including the lack of infection of monocytes and the lack of evidence that the BCR is required for infection of the B cell fraction.  

3. Multiple reviewers raised concerns about the cytokine analysis in primary B cells. These concerns including the design of the assay, the rationale for selection of cytokines to measure, the absence of measurement of additional cytokines, the MOI used, and the inclusion of critical controls to establish the role of the BCR.

4. Modifications to the Introduction, Methods, and Discussion sections should be made, particularly in regard to placing this work in the context of previous findings.

We cannot make any decision about publication until we have seen the revised manuscript and your response to the reviewers' comments. Your revised manuscript is also likely to be sent to reviewers for further evaluation.

Sincerely,

Thomas E. Morrison

Academic Editor

PLOS Pathogens

Alexander Gorbalenya

Section Editor

PLOS Pathogens

Michael Malim

Editor-in-Chief

PLOS Pathogens

orcid.org/0000-0002-7699-2064

Reviewer's Responses to Questions

**Part I - Summary**

Reviewer #1: This manuscript by Gebo and colleagues reports a study on whether BCR on memory B cells that develop after a primary DENV infection can serve as a receptor for secondary DENV infection. The authors postulated that as anti-DENV antibodies can enhance infection via Fc gamma receptors, then BCR activation along with CD79 signaling should functionally be able to serve as an additional entry pathway for DENV during secondary infection. They then showed through HEK293 cells over-expressing transmembrane immunoglobulin that bind DENV that the postulated entry pathway is functional. They also showed that B cells, but not monocytes and T cells, from healthy volunteers who had participated in their DENV-3 challenge studies, were more susceptible to DENV infection compared to non-dengue experienced controls. Finally, they showed that DENV-infected B cells express IL8 although not TNFa, and could potentially thus contribute to the pathogenesis of secondary dengue.

Reviewer #2: The authors of this study aim to prove their excititng and potentially groundbreaking hypothesis that BCR contributes to the susceptibility of B cells and can exacerbate infection in the context of secondary infections by a mechanism they call BDE (BCR-dependent enhancement). They convincingly demonstrated that ectopic expression of a DENV envelope-specific B-cell receptor renders cells susceptible to DENV infection. The authors argued that the frequency of DENV-infectable B cells within an individual could be influenced by their DENV immune status. Furthermore, the study concluded that BCR-dependent infection of B cells leads to the production of proinflammatory cytokines associated with DENV infection in human patients. However, this study did not yet provide convincing evidence for the role of DENV-specific BCR in primary B cells.

Reviewer’s main concerns:

1) Ex vivo secondary heterotypic infection of B cells isolated from volunteers previously infected with DENV 4 did not lead to a substatially higher frequency of DENV+ cells. It is also surprising that monocytes do not become infected, even though myeloid cells are known to be the main viral target in the blood. Can authors explain it? This may be due to the very low MOI used (MOI 0.1) in this important experiment, whereas in Figure 3, an extremely high MOI is chosen. Can the authors explain their choice?

2) It is unclear to this reviewer if, in infection assays, binding or infection is being shown, as there is no UV-infection used as a control. Especially for Figure 2, where the infection frequency in B cells is low, it is crucial to show that this is in fact an active infection.

3) Experiments where BCR is blocked in primary B cells isolated from the DENV-infected cohort would have been crucial to understand the role of BCR in the infection process.

Specific Issues

• The use of MOI 62 in DENV/Ig/BCR experiments is not justified, and it is unclear how these data are relevant to the in vivo situation.

• The purpose of Figure 1G is unclear to this reviewer, 3 points will always yield a high R².

• Based on Figure 2, one could conclude that that none of PBMCs are susceptible to infection as monocytes, which are readily infected in vitro (as many previous studies, some also cited by the authors) are negative here. The reviewer would suggest the use of higher MOI and UV-inactivated virus control.

Reviewer #3: In "B cell receptor dependent enhancement of dengue virus infection", Gebo and colleagues make the intriguing discovery that B-cells are the primary reservoir of dengue virus in patients experiencing acute dengue. This finding poses a potential challenge to the idea that increased severity during secondary dengue is due to ADE in FcG-expressing phagocytic cells. As B cells sustain greater viral loads that phagocytic mononuclear cells, it makes sense that B cells could be the immune cell type that most contributes to secondary infection enhancement and severity. To prove their hypothesis, the authors use flow cytometry to show that cells expressing DENV-specific B-cell receptors were highly susceptible to DENV infection. Furthermore, they showed that the infection-enhancing activity of BCRs correlated with ADE activity of soluble IgGs of the same type. The experiments are mostly well designed, and the findings are novel. The observations in this short manuscript have the potential to change the dogma surrounding secondary DENV infections.

**Part II – Major Issues: Key Experiments Required for Acceptance**

Reviewer #1: The topic of investigation is interesting and important. Studies to understand the elevated risk of severe disease in patients with secondary dengue has thus far focused on antibodies but not BCR on memory B cells. Although the contribution of memory B cell infection to pathogenesis remains unclear, that these cells are infected in patients with secondary dengue has been demonstrated independently by others. Although the study is interesting, a shortcoming of this manuscript is that the data lacks depth. The following points could be considered to strengthen the conclusion:

1. Lines 134-138. To show that BCR that bind DENV could act as a receptor for entry, the authors over-expressed BCR in HEK293 cells. Is there any data on CD79 expression in HEK293 cells? If not, the expressed BCR could merely be binding DENV on the plasma membrane and for it then to move along the membrane to locate clathrin-coated pits for endocytosis. To show that BCR can mediate DENV entry in the same way as Fc gamma receptors, it would be helpful for the authors to consider over-expressing BCR with and without also over-expressing CD79 in HEK293 cells. Alternatively, if HEK293 cells already express CD79, they could consider silencing or knocking out CD79. Without such data, the findings are suggestive but not conclusive of BCR as an entry receptor.

2. Lines 140-143 and supplemental figure 3. The CMV infection rate even in the control is low in Figure 3C. However, the graph in Figure 3D showed a much larger infection rate than that in Figure 3C. Is there an error in one of the figure panels?

3. Lines 168-187. Were the sets of data described in these paragraphs from experiments using DENV-1 RVP or with DENV itself? If so, was it with DENV-4?

4. Lines 185-187. The authors measured IL8 and TNFa. How were these cytokines chosen over others for analysis? If indeed the authors only measured these two cytokines, then perhaps they could take a more untargeted approach to discover if there are other differences in the host response to infection in B cells compared to other cells, such as monocytes. Please also describe the assay/s used in the methods section.

5. The discussion section is too brief. Perhaps the authors would consider discussing how the affinity of binding of DENV to BCR would have on infection since memory B cells would not have undergone as much affinity maturation as plasma cells. Furthermore, they could also discuss the extent in which B cell response to DENV infection contributes to the pathology of dengue, such as vascular leakage and hemorrhage, or that they function merely as trojan horses for DENV to reach myeloid-derived cells, the response of which drives pathology.

Reviewer #2: • The study does not adequately address whether BCR (like FcR in the context of ADE) leads to (productive) infection. It is unclear whether binding or infection occurs, as there is no UV-infection used as a control in the infection assays and the production of new particles has not yet been assessed. Based on Figure 2, it is difficult to conclude if used PBMCs are infectable at all, as primary monocytes, cells that are readily infected in vitro (many previous studies) are negative here regardless of time of their isolation.

• Research has not sufficiently explored whether BCR triggers cytokine production in B cells. It is worth noting that important control conditions for Figure 3C/D, which include DENV + VDB33, DENV + anti-IgG, and VDB3 + anti-IgG, are missing. However, these conditions were featured in the Raji cell experiment, as shown in Figure 3B.

Literature context, previous studies

The introduction and discussion do not elude of any previous research on the processes of direct or antibody-dependent enhancement (ADE) dengue virus (DENV) infection in B cells. One of the studies (referenced by the authors to highlight the importance of B cells) investigated the role of CD300 in primary B cells during direct DENV infection and the inability of these cells to facilitate ADE. However, the text does not address how these findings relate to the authors’ hypotheses, or the new results presented.

Reviewer #3: 1. In Figure 2, the authors show that B cells from patients that had been previously infected with DENV3 where more susceptible to infection with DENV1, while their T cells and monocytes were not. It would be a nice comparison to see how the monocytes of these individuals reacted to DENV1 infection in the presence of their antibodies to compare BDE and ADE activity from the same individual.

2. Because it is difficult to isolate DENV-infectable B cells from DENV-immune individuals, the authors developed a BCR-dependent infection assay to study BCR-mediated pathogenesis. Unfortunately, this crosslinking assay is so far removed from the natural condition that it is hard to agree with the authors’ assertion that “BCR-dependent infection of B cells results in the production of pro-inflammatory cytokines known to be associated with DENV infection in human patients”. Yes, the assay engages BCRs, but these BCRs are from naïve individuals. Perhaps if the assay were better described and discussed, it’s inclusion might be more compelling.

**Part III – Minor Issues: Editorial and Data Presentation Modifications**

Reviewer #1: Line 162. “DENV-infectable B cells” should be “DENV-infected B cells”.

Line 163. I believe this sentence refers to Figures 2C and 2D rather than 2A and 2B.

Reviewer #2: Readability and Clarity

• The text is difficult to read, with abbreviations such as RVP, the types of cells or MOI are not explained in the main text/figures, and only briefly mentioned in the Methods section.

• The Methods section is overly concise, requiring the reader to consult many other articles to understand the basics (e.g., RVP production).

• The number of independent experiments versus technical replicates conducted for most research questions is not specified, with missing N values.

Literature context, previous studies

The introduction and discussion do not elude of any previous research on the processes of direct or antibody-dependent enhancement (ADE) dengue virus (DENV) infection in B cells. One of the studies (referenced by the authors to highlight the importance of B cells) investigated the role of CD300 in primary B cells during direct DENV infection and the inability of these cells to facilitate ADE. However, the text does not address how these findings relate to the authors’ hypotheses, or the new results presented.

Reviewer #3: Minor Comments:

1. Though this is a short report, more detail is required in the methods section. Instead of citing previous work alone, the authors should also give a brief description of what they did in the K562 ADE assays, etc.

2. Figures 3E-G are not described in the text.

PLOS authors have the option to publish the peer review history of their article (what does this mean?). If published, this will include your full peer review and any attached files.

Reviewer #1: No

Reviewer #2: No

Reviewer #3: No
---

## [Decision Letter · Decision Letter 1]

10 Oct 2024

Dear Dr Waickman,

Thank you very much for submitting your manuscript "B cell receptor dependent enhancement of dengue virus infection" for consideration at PLOS Pathogens. As with all papers reviewed by the journal, your manuscript was reviewed by members of the editorial board and by several independent reviewers. The reviewers appreciated the attention to an important topic. Based on the reviews, we are likely to accept this manuscript for publication, providing that you modify the manuscript according to the review recommendations.

The reviewers appreciated the changes made to the revised version of this manuscript and the editors share their enthusiasm. Reviewer 1 and Reviewer 2 identified a few minor concerns including text edits, interpretation of Fig 1G, and some language. We request that the authors review these concerns and revise as needed. We look forward to receiving a revised version of this important study. 

Sincerely,

Thomas E. Morrison

Academic Editor

PLOS Pathogens

Alexander Gorbalenya

Section Editor

PLOS Pathogens

Michael Malim

Editor-in-Chief

PLOS Pathogens

orcid.org/0000-0002-7699-2064

Reviewer Comments (if any, and for reference):

Reviewer's Responses to Questions

**Part I - Summary**

Reviewer #1: This revised manuscript has addressed most of the concerns raised on their original submission. The authors have also included new data showing BDE in memory B cells, ex vivo, from participants of a vaccine study, which has strengthened the authors’ conclusion that BCR can serve as an entry pathway for DENVs.

Reviewer #2: The authors have thoughtfully addressed most of the points raised by the reviewer, and provided additional evidence supporting BCR-mediated infection. The overall conclusion that BCR enhances DENV infection, as stated in the title, abstract and at the end of the introduction, remains unnecessarily overstated. Without data demonstrating BCR's contribution to the production and release of new viruses, the authors should limit their claims to BCR increasing the susceptibility of B cells, rather than suggesting it enhances their permissiveness (i.e., supporting the full viral replication cycle).

Reviewer #3: (No Response)

**Part II – Major Issues: Key Experiments Required for Acceptance**

Reviewer #1: None

Reviewer #2: (No Response)

Reviewer #3: (No Response)

**Part III – Minor Issues: Editorial and Data Presentation Modifications**

Reviewer #1: Several minor considerations remain:

1. Line 140. Technically, as the authors only compared the percentage of dsRed positive cells in the study, the data is not dependent on RVP replication but only translation.

2. Lines 210-212 and Figure 3C,D. The proportion of infected B cells was highest at day 28 compared to both baseline as well as day 90 post-infection. At this early period after convalescence from primary DENV-3 challenge infection, the study participants would likely be refractory to symptomatic secondary DENV infection (Sabin, 1952; Snow et al, Am J Trop Med Hyg 2014). Perhaps this data might suggest that whilst DENV can infect memory B cells via BCR, its role in pathogenesis remains to be determined? Perhaps an inclusion of such a caveat or a more nuanced conclusion would be helpful.

3. Line 250. Typographical error at the end of that line.

Reviewer #2: Ad. 5. (Fig 1G)

The argument that the three data points must be in the same “rank order” to produce a high value reflects a misunderstanding of how the R2 statistic is calculated and interpreted. While rank order may affect the visual representation of data in certain contexts (e.g., when considering monotonic relationships), it is not inherently linked to the calculation of R2 , which measures the proportion of variance in the dependent variable that is explained by the independent variable(s).

To clarify, R2 is based on the degree to which the model’s predictions align with the observed data, calculated as the ratio of the explained variance to the total variance. This means that R2 is determined by the fit of the model, not by the sequence or rank of the data points. For instance, in the case of a simple linear regression with only three data points, a high R2 can occur if the points are near or on a line, irrespective of whether their values increase or decrease in a specific order. The primary factor is the alignment of the model’s predictions with the actual values.

Moreover, with such a small sample size, R2 can become an unreliable measure. Overfitting is a significant risk, as even a poorly generalizable model may fit the three points perfectly, resulting in an artificially inflated . This risk exists regardless of the rank order of the points.

In conclusion, the notion that the rank order of three points influences R2 misunderstands the statistical principles at play. R2 reflects the fit of the model to the data points, not their sequence, and caution should always be exercised when interpreting R2 values derived from small samples. I therefore advise the authors to remove the R2 analysis or interpret it with more caution.

Reviewer #3: (No Response)

PLOS authors have the option to publish the peer review history of their article (what does this mean?). If published, this will include your full peer review and any attached files.

Reviewer #1: No

Reviewer #2: No

Reviewer #3: No

Figure Files:

Data Requirements:

Reproducibility:

References:

---

## [Editor Report · Decision Letter 2]

18 Oct 2024

Dear Dr Waickman,

We are pleased to inform you that your manuscript 'B cell receptor dependent enhancement of dengue virus infection' has been provisionally accepted for publication in PLOS Pathogens.

Best regards,

Thomas E. Morrison

Academic Editor

PLOS Pathogens

Alexander Gorbalenya

Section Editor

PLOS Pathogens

Michael Malim

Editor-in-Chief

PLOS Pathogens

orcid.org/0000-0002-7699-2064
---

## [Editor Report · Acceptance letter]

24 Oct 2024

Dear Dr Waickman,

We are delighted to inform you that your manuscript, "B cell receptor dependent enhancement of dengue virus infection," has been formally accepted for publication in PLOS Pathogens.

Best regards,

Michael Malim

Editor-in-Chief

PLOS Pathogens

orcid.org/0000-0002-7699-2064